# An Efficient Approach to Predict Eye Diseases from Symptoms Using Machine Learning and Ranker-Based Feature Selection Methods

**DOI:** 10.3390/bioengineering10010025

**Published:** 2022-12-24

**Authors:** Ahmed Al Marouf, Md Mozaharul Mottalib, Reda Alhajj, Jon Rokne, Omar Jafarullah

**Affiliations:** 1Department of Computer Science, University of Calgary, Calgary, AB T2N 1N4, Canada; 2Department of Computer Science and Engineering, Daffodil International University, Dhaka 1341, Bangladesh; 3Department of Computer and Information Sciences, University of Delaware, Newark, DE 19716, USA; 4Department of Computer Engineering, Istanbul Medipol University, Istanbul 34810, Turkey; 5Department of Heath Informatics, University of Southern Denmark, 5230 Odense, Denmark; 6Ispahani Islamia Eye Hospital, Dhaka 1215, Bangladesh

**Keywords:** eye disease, machine learning, ranker-based feature selection, symptomatic analysis, support vector machine

## Abstract

The eye is generally considered to be the most important sensory organ of humans. Diseases and other degenerative conditions of the eye are therefore of great concern as they affect the function of this vital organ. With proper early diagnosis by experts and with optimal use of medicines and surgical techniques, these diseases or conditions can in many cases be either cured or greatly mitigated. Experts that perform the diagnosis are in high demand and their services are expensive, hence the appropriate identification of the cause of vision problems is either postponed or not done at all such that corrective measures are either not done or done too late. An efficient model to predict eye diseases using machine learning (ML) and ranker-based feature selection (r-FS) methods is therefore proposed which will aid in obtaining a correct diagnosis. The aim of this model is to automatically predict one or more of five common eye diseases namely, Cataracts (CT), Acute Angle-Closure Glaucoma (AACG), Primary Congenital Glaucoma (PCG), Exophthalmos or Bulging Eyes (BE) and Ocular Hypertension (OH). We have used efficient data collection methods, data annotations by professional ophthalmologists, applied five different feature selection methods, two types of data splitting techniques (train-test and stratified k-fold cross validation), and applied nine ML methods for the overall prediction approach. While applying ML methods, we have chosen suitable classic ML methods, such as Decision Tree (DT), Random Forest (RF), Naive Bayes (NB), AdaBoost (AB), Logistic Regression (LR), k-Nearest Neighbour (k-NN), Bagging (Bg), Boosting (BS) and Support Vector Machine (SVM). We have performed a symptomatic analysis of the prominent symptoms of each of the five eye diseases. The results of the analysis and comparison between methods are shown separately. While comparing the methods, we have adopted traditional performance indices, such as accuracy, precision, sensitivity, F1-Score, etc. Finally, SVM outperformed other models obtaining the highest accuracy of 99.11% for 10-fold cross-validation and LR obtained 98.58% for the split ratio of 80:20.

## 1. Introduction

The eye is considered to be the most important sensory organ for humans and it plays a vital role in the overall ability of humans to interact with the world. Unfortunately many individuals in both rural and urban areas suffer from eye conditions such as cataracts, glaucoma, ocular hypertension, bulgy vision, etc. that affected their vision. There are a variety of causes for this such as age, diabetes, genetic and inheritance. Modern lifestyles, which have led to increased use of displays for digital devices, are also a factor affecting the vision.

These eye problems have a particularly high occurrence in many South Asian countries. As an example, in Bangladesh 1.5% of the adult population is blind and 21.6% of the population has low vision according to a survey presented in [1]. There are various reasons for this including the lack of vision care by individuals, pollution and excessive use of digital devices [1].

We chose individuals assumed to have one of the five eye diseases that are currently prevalent in Bangladesh and collected datafor the diseases (biomarkers and disease symptoms). These data are very important for any ophthalmologist or clinical professional since it is used to guide the treatment of the disease. It can also be used to train artificial intelligence algorithms that can ensure that the correct disease has indeed be diagnosed and provide automated recommendations for how to treat the disease.

From a literature search one finds that classic machine learning algorithms, especially classification algorithms, are the most appropriate for the detection and recommendation of the five eye diseases considered in this paper [2,3,4].

A dataset of eye-disorder-related data was compiled. This dataset will be used to make the eye disease classification easier and and it may also be used for further studies of eye diseases. Multiple machine learning techniques were applied to this dataset to test its applicability to the detection and classification tasks considered in this paper. The dataset can also be used to test other platforms such as Chabot, web apps, etc. which in turn can help the world with more accurate information about eye disease issues. One of the goals in this paper is also to make relevant and accurate medical information about eye diseases available to the medical community.

In the case of machine learning algorithms, it is also important that predictions can be explained since this provides insights into the predicted phenomena. Hence the explainable artificial intelligence (XAI) methods used here are of great help for the understanding of the eye disease symptoms for a given disease prediction.

The main contributions of this study are:Creating a benchmark dataset in the domain of eye diseases validated by professional ophthalmologists, that cam be applied to test ML, AI and Symptomatic analyses.Utilizing ranker-based feature selection methods to identify highly ranked symptoms among the five diseases.Experimenting with scenarios both with and without splitting the dataset and with several feature selection methods for better predictions.Compare the performance of classic ML methods to efficiently predict the occurrence of eye diseases.

For the remainder of the paper, Section 2 provides an overview of the five eye diseases that are considered in this study. Section 3 describes the existing related studies performed on these diseases. The proposed methodology along with detailed descriptions of each of the methodological steps are described in Section 4. The measurement indices used to measure the performance of the applied machine learning models are given in Section 5 and the experimental results are discussed in Section 6. A constructive discussion regarding the results and comparison of the model performances are presented in Section 7, and finally Section 8 concludes with suggestions for future improvements.

## 2. Overview of Five Eye Diseases

In this section, a brief overview of the five eye diseases that we have worked on is presented. For each disease, we describe the disease conditions, disease symptoms and disease risk factors. The images of the selected five categories of eye diseases are shown in Figure 1, however we have not used image data to identify or to annotate the dataset. Therefore, the sources or the images are referenced in the footnotes. The next five subsections describe the diseases worked on.

### 2.1. Cataracts

A dense and cloudy area that is usually formed on the lens of the eye is called a cataract. This cataract is an agglomeration of proteins situated in the eye that forms a lump-like mass which prevent the normal transmission of light through the lens to the retina. Some of the major symptoms appearing due to cataracts include degradation of night vision, blurry vision, faded colors and double vision. The different sub-types of cataracts are nuclear, cortical, posterior, congenital, senile, traumatic, bilateral as well as other subtypes. These sub-types are grouped into one disease category for this study to minimize the complexity of asociating the cataract with a specific sub-type. Smoking, older age, obesity, increased blood pressure, history of the same disease in family members, diabetes, exposure to radiation from X-ray and cancer treatments are the main risk factors associated with cataracts [5]. Cataracts are one of the common eye diseases in low and middle-income countries [6].

### 2.2. Acute Angle-Closure Glaucoma (AACG)

AACG is caused by a sudden increase in the intra-ocular pressure (IOP) which causes a displacement of the apposition of the iris to the trabecular meshwork. In acute angle-closure (AAC), the common symptoms are blurred vision, red eye, pain, headache, nausea and vomiting [7]. The process of AAC occurs suddenly with a dramatic onset of symptoms. Because of premorbid comorbidities, severely ill patients may encounter the risk of a sight-threatening condition. The drainage of the aqueous humor may be restricted causing high interior pressure that can result in critical damage to the optic nerve [7].

### 2.3. Primary Congenital Glaucoma (PCG)

PCG is another type of glaucoma caused by damage to the optic nerve [8]. High intra-ocular pressure in the eye is the main cause of the damage. It is identified when excessive tearing, light sensitivity, hazy cornea, redness of the eye, the closing of the eyelid and cloudy cornea symptoms are present. For this study, data for PGG patients are separated from the data for Secondary Congenital Glaucoma (SCG) patients. Having a previous related medical history in the family is considered one of the risk factors for PCG [9].

### 2.4. Exophthalmos or Bulging Eyes

Bulging eyes (BE) which is also known as exophthalmos, the medical term for BE. It is not a condition by itself, but a symptom of other conditions. Symptoms of BE include excessive dryness, visible whiteness, double vision and throbbing sensation in the eyes [10]. One or both eyes of a patient can be affected by BE and when it is present it may cause blinking problems.

### 2.5. Ocular Hypertension

Ocular hypertension (OH) is caused by poor drainage of the aqueous humor, the fluid inside the eye. Risk factors for developing OH are patients having a previous family history of ocular hypertension, glaucoma, diabetes, and age over 40. Patients having unbalanced diabetes or high blood pressure may be exposed to ocular hypertension [11].

## 3. Related Works

Papers relating to the prediction of eye diseases using artificial intelligence and machine learning are reviewed in this section since these areas discuss the tools that are used for the proposed method for efficiently predicting the five eye diseases discussed above. Some papers discussing explainable artificial intelligence methods are also included since some results from these paper are used for better interpretation of the results from the proposed method.

The few works that have been performed in the area of eye disease prediction using symptomatic data are also considered here.

One study assessed the prevalence of eye diseases in a certain low-income community in Bangladesh. The study was performed using in-person surveys and the results showed a higher prevalence of eye diseases than the world average. This reflects the difference in eye disease prevalence and diagnosis between individuals in low-income individuals and individuals with higher standards of living [1]. The study found that there is a funding and communication gap between vision related support for slum dwellers of Bangladesh and the Bangladesh community in general.

Ref. [12] is an epidemiology study of eye diseases that targets the Shahjadpur Children’s Cohort (SCC), a very interesting population-based cohort including only children with disabilities [12]. A chi-squared test, Fisher’s exact test and the binomial test were performed to find significant variations among the SCC. Sociodemographic details and the prevalence of childhood disabilities in the cohort study were included in the study.

Kadir et al. [13] performed a cross-sectional study of vision among 252 primary school-going children of the southern region of Bangladesh. The study was performed to assess the refractive errors (Myopia, Ametropia and Hyperopia) in the cohort and it was found that myopia is a common occurrence (50%).

We found some articles presenting methodologies to address eye disease classification using image processing techniques and ML algorithms. For these methodologies it is important for classification to have a large enough set of image data to run the models with image processing abilities that detect the region of interest (ROI) in the image. One article, Sakri et al. [14], presented an automated classification framework using image processing to classify diabetic eye disease (DED). Image enhancement techniques such as contrast limited adaptive histogram equalization (CLAHE), illumination correction and image segmentation techniques such as finding blood vessels, macular region and optic nerve were applied before applying the pre-trained model (VGG-16, Xception, ResNet50, CNN). All the models performed well with accuracies over 90% correct recognition. Similar image processing-based articles with different datasets can be found in [15,16,17,18].

Neural network based approach using the image dataset is a very common scenario in detecting eye diseases. Nazir et al. [19], extracted features using DenseNet-100 and applied improved CenterNet method on Aptos-2019 and IDrID dataset. The highest accuracy obtained in this method are 97.93% using the Aptos-2019 dataset and 98.10% using the IDrID dataset. The same dataset of Aptos-2019 has been used in [20], with feature fusion techniques and a deep neural network. The accuracy obtained in this method is 84.31%. Khan et al. [21] tried to manually extracted the retinal features with no feature-selection algorithms applied. The combination of CNN with VGG-19 has been proposed in this paper and accuracy obtained is 97.47%. Sarki et al. [22] and Pahuja et al. [23] also applied CNN for the image datasets and obtained accuracy less thatn 90% in both the case.

A data-driven approach for eye disease classification was adopted by Malik et al. [24]. The authors prepared a dataset having ten (10) attributes including age, gender and complaint (pain or blurred vision). Visual acuity of the right and left eye, pinhole value of the left and right eye, symptoms obtained from a slit lamp test (lids, upper lid, site, swelling), and posterior segment test are the overall symptoms considered in this study to determine the diagnosis glaucoma vs unspecified primary angle-closure glaucoma. NB (81.53%), DT (85.81%), Neural Network (86.98%) and RF (86.63%) models were applied to predict the diagnosis, and neural networks outperformed the other models.

A recent work [25] presented a multi-categorical common eye disease detection method using CNN. They prepared their own hand-crafted dataset which had 2250 images of cataracts, conjunctivitis and normal eyes. Three transfer learning models (VGG-16, Resnet-50 and Inception-v3) were applied, where the class-wise performance evaluation was shown for each model. Inception-v3 (97.08%) outperformed VGG-16 (95.48%) and Resnet-50 (95.78%) in the experiments and they claimed the results to be the highest among the other image-based eye disease classification approaches [26,27,28,29].

From the review of the existing works, we found a gap in finding the most relevant symptoms for different types of eye diseases as well as the lack of a benchmark dataset with symptomatic data of eye diseases. In this study, we have focused on these gaps and we have designed a research methodology to fill the gaps.

## 4. Research Methodology

An overview of the research methodology adopted for eye disease predictions in this study is shown diagrammatically in Figure 2. The steps of the process are highlighted, starting from the data collection from patients to the application of ML and XAI methods. The remainder of this section elaborates on the steps of the proposed methodology.

### 4.1. Data Collection

Data collection processes have in general to be executed rigorously in order to create a valid and useful dataset. The data collected has to include both symptoms and biomarker data for each patient included in the dataset. For this study, we have included real-world patients data collected when they were present during a referral to an ophthalmologist. Since it can be very difficult to track the symptoms by the ophthalmologist during a short visit, there were two interviewers (ophthalmologist and one of the author) collecting the data. The statistical properties of the dataset acquired are presented in Table 1.

The patients were examined by a practicing ophthalmologist while checking the conditions and biomarkers listed in Table 2 for each of the patients. The dataset collected contains 563 patients data having any one of the eye diseases mentioned in Section 2. The data collection was also conducted in a closed-room environment during one-to-one appointments and the attributes noted in Table 2 were collected. A value, either 0 or 1 has been assigned for each attributes based on the observation of the ophthalmologist to avoid a missing value in the dataset. This makes it a robust dataset having no missing values.

### 4.2. Data Annotation

Data annotation or labeling of the dataset used for the training of a methodology that is based on supervised machine learning algorithms is the most critical and important step in the development of the methodology. If the annotation or labeling of the data is not performed properly then the acceptability of the whole methodology might be questioned. In this case, the data collection was performed very carefully. While collecting the data, the ophthalmologist played the role of an interviewer and the patients played the role of an interviewee. Questions associated with the 19 attributes were asked and properly examined by the ophthalmologist to determine the right kind of eye disease that the patient had. Hence, the ophthalmologist worked as the annotator of the dataset. Therefore, the dataset created for the study has been validated by domain experts, hence it can be considered as a benchmark dataset in this domain. Each attribute is labeled either zero (0) or one (1) based on the examination performed by the ophthalmologist. The annotated dataset contains 19 attributes/symptoms and one (1) class label indicating the eye disease. This annotated dataset was forwarded to the next step in the feature selection process.

### 4.3. Feature Selection Methods

For the feature selection step, we considered ranker-based feature selection methods, so that we got a list of attributes, in this case, symptoms. It is expected that this selected symptom list may be significant for predicting a particular eye disease more efficiently.

The ranker-based methods utilized in this study are Pearson Correlation (PC), Information Gain (IG), Principal Component Analysis (PCA), Relief-based Ranking (RR) and all features. The methods were applied directly to the annotated data and the ranking scores of each attribute were determined. Brief descriptions of the feature selection methods are given next.

The Pearson Correlation (PC)-based feature selection [30] method is widely used in machine learning problems. The ($\rho_{X,Y}$) is calculated using Equation (Equation 1) where cov(X,Y) is the covariance between the *X* and *Y* and σ the standard deviations (SDs) on the *X* and *Y*:(1)$$\rho_{X,Y}=\frac{cov(X,Y)}{\sigma_{X}\sigma_{Y}}$$

The *X* and *Y* can be considered as class-feature or feature-feature relationships. *X* can be the class and *Y* can be the feature or *X* can be the feature and *Y* can be another feature. The $\rho_{X,Y}$ value is between the −1 and +1, where −1 means a negative correlation between *X* and *Y* where 0 means no correlation between *X* and *Y* and +1 means a positive correlation between *X* and *Y*. The higher the $\rho_{X,Y}$-value the higher the correlation between *X* and *Y* is. Therefore, for our study, we have chosen the class-feature relationship. We calculated the class-feature correlation values for all the features and ranked them by the correlation values from high to low. Depending on the ranking, we selected some of the features to run further steps of the ML models.

Information Gain (IG) is another widely accepted feature selection method for various research problems for example where text categorization is used [31,32]. This is evidenced by the use of IG in several research domains, such as computer vision [32] and text classification [33]. IG is a ratio value calculated by Equation (Equation 2).
(2)$$IG\left(T,a\right)=H\left(T\right)-\sum_{v\in values(a)}^{}\frac{|x\in T|x_{a}=v|}{\left|T\right|}\cdot H\left(x\in T|x_{a}=v\right)$$

Here, values(a) is the set of all possible values of features a∈Attr where Attr is the set of all features, *H* is the entropy, and x∈T denotes the value of specific example *x* for a∈Attr. The largest IG is the smallest entropy.

Principal Component Analysis (PCA) was invented by Karl Pearson, see [34,35], initially as an analog of the principal axis theorem in mechanics. After the development of eigenvalue decomposition and other related theorems, the use of PCA became more popular. PCA is the method for calculating the principal components of a dataset. This multivariate technique tries to analyze the data in which the observations are kept by several inter-correlated quantitative variables, which are dependent [36]. PCA can be used as a method to reduce the dimension of the data to handle high dimension data for a given process [37]. The dimensionality reduction is performed by choosing an optimal number of eigenvectors to account for some percentage of variance in the original data.

Relief is a filter-based feature selection algorithm presented by Kira and Rendell in 1992 [38]. The algorithm takes a data set with *p* instances of *n* number of features. The method iterated *m* times starting with a *n*-long weight vector *W*. In each iteration the weight vector is updated as in Equation (Equation 3).
(3)$$W_{i}=W_{i}-{\left(x_{i}-nearHit_{i}\right)}^{2}+{\left(x_{i}-nearMiss_{i}\right)}^{2}$$

For each iteration, the feature vector (*X*) is assigned to one random instance, and the closest same-class instance is called ‘near-hit’. Similarly, the closest different-class instance is called ‘near-miss’. Therefore, the weight of any given attribute will decrease if it differs from that feature and increase in the reverse case.

An associated ranker method ranking the features by their individual evaluators (Correlation, GainRatio, Components and Relief) was implemented with the feature selection methods. The method may choose a specific number of features to be retained for the remainder of the process. We have kept the default value (−1) to find the ranking over all the features. No initial set of features was chosen to bias the whole process. Therefore, all of the features are considered as initial inputs to the methods.

### 4.4. Data Splitting Strategies

For the ML algorithms, especially when applying classifiers, it is very important to perform data splitting for the train-test mechanism [39]. One part of the data is used for training the models and the rest of the data are used for testing the performance of the model. Therefore, the choice of what percentage should be used for training and what percentage should be used for testing can be critical. In this study, we adopted two data-splitting strategies: Train-Test and k-fold Cross Validation.

For the Train(%)-Test(%) technique, we split the data using 66–34%, 75–25% and 80–20%. Using the different splits we ran the same models on the data, to find the performance of the ML algorithm.

The second data splitting strategy is k-fold Cross Validation [40]. Though it is a kind of sampling method, it has been found to be effective in the area of ML while doing the Train–Test split. In a single fold of cross-validation, the data are partitioned into two parts (training and testing) and in the second fold, the same data are partitioned randomly. In this paper, k-fold was used, which means that the original data sample was randomly partitioned into *k* equal-sized sub-samples. Among those *k* sub-samples, one sub-sample was considered for testing and the rest of the sub-samples were considered for training and the same process was run *k*-times. In the same manner, as for the Train-Test strategy, we chose 3-fold, 5-fold and 10-fold Cross Validation for finding the best-performing ML algorithm.

### 4.5. Machine Learning Methods

The naive Bayes classifier simplifies the classifying process considerably by assuming that the presence of a particular feature in a class is not related to any other feature in the class [41]. Although this independence is generally a poor assumption, in practice naive Bayes often competes well with more sophisticated classifiers [41]. Our broad goal was to understand the data characteristics which affect the performance of naive Bayes [41]. Our approach used Monte Carlo simulations that allow a systematic study of classification accuracy for several classes of randomly generated problems [41]. The success of naive Bayes in the presence of feature dependencies can be explained as follows: optimality in terms of zero-one loss (classification error) is not necessarily related to the quality of the fit to a probability distribution (i.e., the appropriateness of the independence assumption). Rather, an optimal classifier is obtained as long as both the actual and estimated distributions agree on the most probable class [41]. For example, naive Bayes optimality can be proven for some problem classes that have a high degree of feature dependencies, such as disjunctive and conjunctive concepts [41].

The k Nearest Neighbor (k-NN) method is a popular classification method in data mining and statistics because of its simple implementation and excellent classification performance [42]. However, it is impractical for traditional k-NN methods to assign a fixed k value (even if it is set by experts) to all test samples [42]. Previous solutions assigned different k values to different test samples by the cross-validation method but this was usually time-consuming [42]. This paper proposes a k-Tree method to learn different optimal k values for different test/new samples, by involving a training stage in the k-NN classification [42].

For a simplified description, decision tree analysis is a divide-and-conquer approach to classification (and regression which is not covered within the scope of this review) [43]. Decision trees can be used to discover features and extract patterns in large databases that are important for discrimination and predictive modeling [43]. These characteristics, coupled with their intuitive interpretation, have been some of the reasons for the extensive use of decision trees for both exploratory data analysis and predictive modeling applications for more than two decades [43]. Decision trees have an established foundation in both the machine learning and artificial intelligence literature and a niche in the use of decision trees in both the chemical and biochemical sciences is slowly developing [43].

In the same manner, as contingency table analyses and two tests, Logistic Regression (LG) allows the analysis of dichotomous or binary outcomes with two mutually exclusive levels and it allows the use of continuous or categorical predictors and provides the means for adjusting for multiple predictors [44]. This makes LG especially useful for the analysis of observational data when adjustments are needed to reduce the potential bias resulting from differences in the groups being compared [44].

## 5. Performance Measurement Indices

We adapted the widely accepted measurement tools accuracy, precision, recall and F1-score for evaluating the performance of the applied ML methods. Usage of these measurement indices can be found in many existing works, including [45,46]. While calculating these indices, the positive or negative classification of the diseases is taken into account. The following Equations (Equation 4)–(Equation 7) were used for generating the measurements.
(4)$$Accuracy\left(ACC\right)=\frac{TP+TN}{TP+TN+FP+FN}$$
(5)$$Precision=\frac{TP}{TP+FP}$$
(6)$$Sensitivity\left(SEN\right)=\frac{TP}{TP+FN}$$
(7)$$F1-score=\frac{2(Precision\times Recall)}{\left(Precision+Recall\right)}$$

Here, TP is True Positive (when the ML model correctly classifies a patient as having a particular eye disease), TN is True Negative (when the ML model correctly classifies a patient as having a different eye disease), FP is False Positive (when the ML model incorrectly classifies a patient as having one particular disease when the patient actually has another disease) and FN is False Negative (when the Ml model incorrectly classifies a patient as not having a disease when the patient actually has the disease).

## 6. Experimental Results

In this section, we show the detailed results of the experiments. After applying the feature selection methods the selected features along with their ranking values are given in this section. The experiments were executed based on splitting and feature selection. The performance measurement indices mentioned in Section 5 were used to describe the outcomes of the experiments.

### 6.1. Applying Feature Selection Methods

We applied multiple ranker-based feature selection techniques and the ranking score of the features is therefore important when choosing the high-scored features for the further process. The ranking score, attribute names and numbers are shown for PC, IG and RR in Table 3, Table 4 and Table 5, respectively. PCA on the other hand gives a ranking with associated attributes for obtaining the best results based on the outcome or class. Therefore, it only ranked the first sixteen (16) features, as shown in Table 6. After applying the FS methods, the first ten attributes were selected and considered for the next steps.

### 6.2. Experiments on Data Splitting and FS Methods

We devised four (4) experiments to test multiple data-splitting strategies and feature selection techniques. All of the ML methods were applied in these experiments. For the first two experiments, we considered splitting with and without the feature selection applied. For the next two experiments, cross-validation was applied with and without feature selection methods. As described in Section 4, subsection D, 66–34%, 75–25% and 80–20% are the splitting criteria used and 3-fold, 5-fold and 10-fold cross-validations are applied.

#### 6.2.1. Experiment-1: Splitting + FS Applied

Data splitting was performed and five feature selections were applied for this experiment. The effect of the selection features can be found in this experiment. Comparison of precision and recall/sensitivity values are shown in Figure 3.

The accuracy values for each of the models for different train-test split scenarios are shown in Figure 4 and Figure 5. The highest accuracy obtained for XGBoost is 98.23%.

#### 6.2.2. Experiment-2: Splitting + No FS Applied

For experiment-2, splitting was applied, but the FS methods were not applied. Precision, recall, F1-score and accuracy of the ML models are shown in Table 6. XGBoost performs better overall showing more than 98% accuracy for all types of splits. The highest accuracy reported was 98.582% when LR was used in a 75–25% split.

#### 6.2.3. Experiment-3: Cross-Validation + FS Applied

In experiment-3, we applied a cross-validation technique instead of a split with the application of five feature selection strategies. This resulted in a total of one hundred and thirty-five (135) runs of the ML models with different setups. This rigorous experiment gave us the most suitable ML model among all the models chosen for the study. The precision, recall, F1-score and accuracy values are presented in Table 7. Among all the models, LR outperformed other models showing 98.94% accuracy.

#### 6.2.4. Experiment-4: Cross-Validation + No FS Applied

In experiment-4, we applied three different cross-validation methods, but this time without selecting any particular features coming from the FS methods. This method can be easily compared to experiment-2 to compare the percentage split and cross-validation methods. Table 8 shows the precision, sensitivity, F1-score and accuracy values for each of the ML models. SVM showed the highest accuracy of 99.11% in this experimental setup. Table 9 shows the same measurements as Table 8 for cross-validation without the FS methods. And SVM outperforms the other algorithms in 10-fold cross-validation obtaining 99.110% accuracy.

## 7. Discussion

### 7.1. Finding Significant Features

The list of significant features based on the feature selection algorithms is depicted in Table 3, Table 4, Table 5 and Table 6. The selected features or attributes can be considered better choices among the other symptoms of eye disease. The first ten selected features can be formulated as four different feature sets outputted from PC-based ($F_{PC}$), IG-based ($F_{IG}$), PCA-based ($F_{PCA}$), and Relief-based ($F_{Relief}$) feature selection methods, as below.
(8)$$F_{PC}=<a1,a2,a3,a5,a7,a8,a9,a11,a12,a15>$$
(9)$$F_{IG}=<a1,a2,a3,a7,a8,a9,a11,a12,a13,a15>$$
(10)$$F_{PCA}=<a1,a2,a3,a4,a10,a13,a14,a16,a17,a18>$$
(11)$$F_{Relief}=<a1,a2,a3,a7,a8,a9,a11,a12,a15,a18>$$

### 7.2. Finding Common Features

Considering the selected feature sets in Equations (Equation 8)–(Equation 11), the common set can be found calculating the intersection of the features.
(12)$$F_{PC}\cap F_{IG}=<a1,a2,a3,a7,a8,a9,a11,a12,a13>$$
(13)$$F_{PC}\cap F_{PCA}=<a1,a2,a3>$$
(14)$$F_{PC}\cap F_{Relief}=<a1,a2,a3,a7,a8,a9,a11,a12,a13>$$
(15)$$F_{IG}\cap F_{PCA}=<a1,a2,a3,a13>$$
(16)$$F_{IG}\cap F_{Relief}=<a1,a2,a3,a7,a8,a9,a11,a12,a13>$$
(17)$$F_{PCA}\cap F_{Relief}=<a1,a2,a3>$$
(18)$$F_{PC}\cap F_{IG}\cap F_{PCA}\cap F_{Relief}=<a1,a2,a3>$$

Considering $F_{PC}$ as A, $F_{IG}$ as B, $F_{PCA}$ as C and $F_{Relief}$ as D, the Venn diagram in Figure 6 shows the common features among the sets. The numbers inside the elipses shows the number of features they have in common. For example, AB (9) means there are 3 common features between A and B. Similarly, ABCD (3) means there are 3 common features between the A, B, C, and D sets.

### 7.3. Comparison with Existing Works

In this section, we have compared our proposed work with the existing works on the same domain focusing on the prediction of eye disease. Most of the works from literature, used image data to detect eye disease. The images contain the disease region of interest (RoI) and it is easier to identify the RoI by the experienced ophthalmologists. The challenge of annotation can be handled easily by the domain experts and then the supervised learning algorithms would be appropriate for the prediction. The recent advancements of machine learning and deep learning methods also encouraged the researchers to exploit them for the eye disease detection. The following Table 10 shows the comparison with the recent existing works from the literature with our proposed approach. None of the approaches mentioned in Table 10 have applied feature selection algorithms to find out the significant symptom features. We have applied several feature selection methods to find out the significant and common features affecting the organ. Our proposed approach obtained highest accuracy 99.11% (SVM) by exploring the classical ML methods.

## 8. Conclusions

This paper presents an efficient approach for predicting five different eye diseases and shows a comparative analysis among the ML methods for predicting the five common eye diseases from a benchmark dataset. The most critical issue with suitable dataset for ML models is annotation or class labelling. For our work, we have annotated the data by practicing ophthalmologist, which gives more accurate and validate data for the models. Therefore, the ML models are showing very satisfactory results in terms of accuracy values, as most of then showing accuracy above 90% and all the models are showing accuracy over 70%. The highest accuracy obtained is 99.11% from SVM with cross-validation and without applying any feature selection methods. The significant features are identified using feature selection methods and the intersection of the selected features are showing the common features. From the common features we obtain the understanding of the symptoms responsible for the eye diseases. One of the shortcomings of the paper is that we have not used any images for the predictive analysis. Using image data along with the annotated symptom data would have provide better solution to a multi-model approach. In future works, a multivariate or uni-variate analysis may be conducted to identify specific symptoms and acquire insights about a particular eye disease. The application of explainable artificial intelligence to interpret the best model could be another improvement of this work. 

## Figures and Tables

**Figure 1 bioengineering-10-00025-f001:**
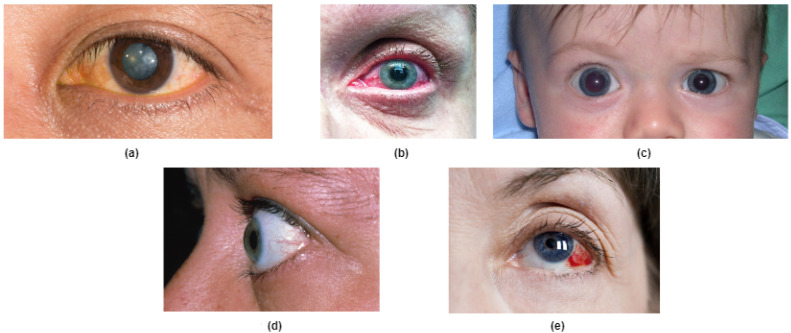
Typical images of eye diseases. (**a**) Cataracts, (**b**) Acute angle-closure glaucoma, (**c**) Primary congenital glaucoma, (**d**) Exophthalmos or bulging eyes, and (**e**) Ocular hypertension.

**Figure 2 bioengineering-10-00025-f002:**
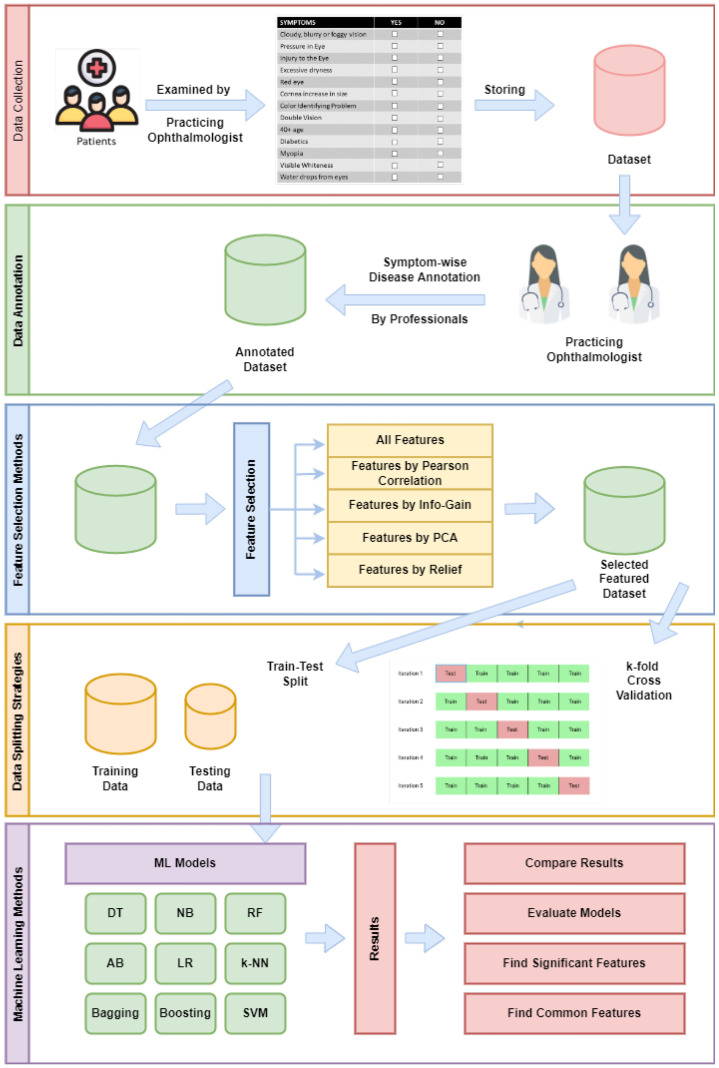
Overview of working diagram of the proposed eye disease prediction method.

**Figure 3 bioengineering-10-00025-f003:**
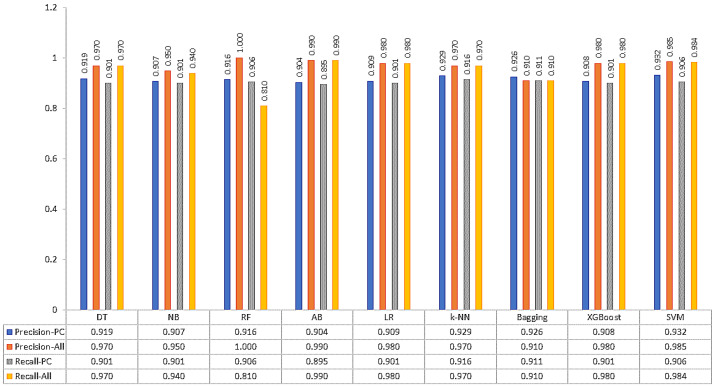
Precision and Recall values of ML methods with 66–34% split and PC-based FS method.

**Figure 4 bioengineering-10-00025-f004:**
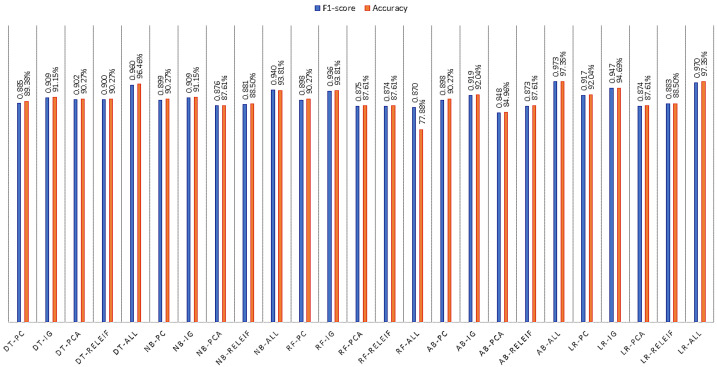
F1-score and Accuracy values of DT, NB, RF, AB and LR with 80%–20% split and five FS method.

**Figure 5 bioengineering-10-00025-f005:**
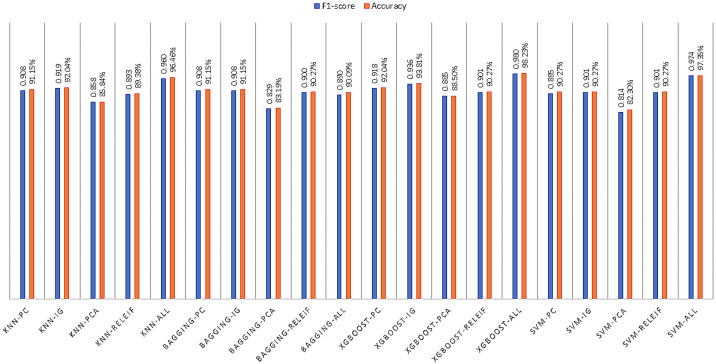
F1-score and Accuracy values of k-NN, Bagging, XGBoost and SVM with 80–20% split and five FS method.

**Figure 6 bioengineering-10-00025-f006:**
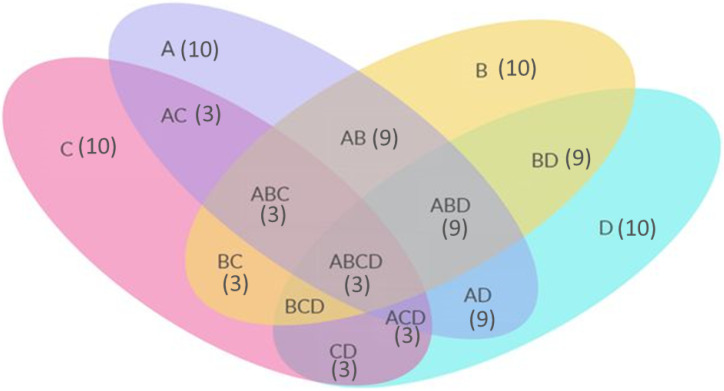
Venn Diagram showing the common features among the feature sets.

**Table 1 bioengineering-10-00025-t001:** Statistical Properties of Dataset.

Properties	Amount/Values
No. of patients	563
Age group of patients	23–65
Gender of patients	Male or Female
No. of instances in the dataset	563
Data collection process	In-person interview with patients
Type of interview	Semi-structured interviews
Type of pre-defined questionnaire	Binary closed questions (Yes/No)
Types of eye diseases	Cataracts (236 instances) Acute Angle-Closure Glaucoma (AACG) (59 instances) Primary Congenital Glaucoma (PCG) (57 instances) Exophthalmos/Bulging Eyes (BE) (41 instances) Ocular Hypertension (170 instances)

**Table 2 bioengineering-10-00025-t002:** Attributes used for data collection.

No.	Attributes	Properties
a1	Cloudy, blurry or foggy vision	The values are either 0 or 1
a2	Pressure in eye	
a3	Injury to the eye	
a4	Excessive dryness	
a5	Red eye	
a6	Cornea increased in size	
a7	Problem in identifying color	
a8	Double vision	
a9	Myopia	
a10	Trouble with glasses	
a11	Hard to see in the dark	
a12	Visible whiteness	
a13	Mass pain	
a14	Vomiting	
a15	Water drops from eyes continuously	
a16	Presence of light when eye lid closes	
a17	Family history of similar disease	
a18	Age +40	Biomarker (0 or 1)
a19	Diabetes	

**Table 3 bioengineering-10-00025-t003:** Ranking of Features and associated correlation coefficient values from PC-based FS method.

Ranking Score	Attributes (Attribute Number)
0.6218	Cloudy, blurry or foggy vision (a1)
0.5583	Problem in identifying color (a7)
0.5340	Double vision (a8)
0.5148	Water drops from eyes continuously (a15)
0.5114	Pressure in eye (a2)
0.5073	Hard to see in the dark (a11)
0.4991	Myopia (a9)
0.3052	Injury to the eye (a3)
0.2403	Mass pain (a13),
0.2279	Red eye (a5)
0.2258	Vomiting (a14)
0.2225	Cornea increased in size (a6)
0.2225	Presents of light when eyelid close (a16)
0.2061	Visible whiteness (a12)
0.2023	Excessive dryness (a4)
0.1539	40+ Age (a18)
0.1531	Family history of similar disease (a17)
0.1525	Diabetes (a19)
0.0524	Trouble with glasses (a10)

**Table 4 bioengineering-10-00025-t004:** Ranking of Features from Information Gain-based FS method.

Ranking Score	Attributes (Attribute Number)
0.8000	Cloudy, blurry or foggy vision (a1)
0.6182	Problem in identifying color (a7)
0.5587	Water drops from eyes continuously (a15)
0.5434	Double vision (a8)
0.5430	Pressure in eye (a2)
0.5216	Myopia (a9)
0.4848	Hard to see in the dark (a11)
0.3477	Injury to the eye (a3)
0.2326	Mass pain (a13)
0.2293	Visible whiteness (a12)
0.2196	Excessive dryness (a4)
0.2130	Cornea increased in size (a6)
0.2130	Presents of light when eyelid close (a16)
0.2108	Red eye (a5)
0.2004	Vomiting (a14)
0.0749	40+ Age (a18)
0.0730	Family history of similar disease (a17)
0.0718	Diabetes (a19)
0.0000	Trouble with glasses (a10)

**Table 5 bioengineering-10-00025-t005:** Ranking of Features from Relief-based FS method.

Ranking Score	Attributes (Attribute Number)
0.6194	Cloudy, blurry or foggy vision (a1)
0.4196	Pressure in eye (a2)
0.4181	Problem in identifying color (a7)
0.3682	Injury to the eye (a3)
0.3383	Myopia (a9)
0.3253	Double vision (a8)
0.3186	Water drops from eyes continuously (a15)
0.3137	Hard to see in the dark (a11)
0.1129	40+ Age (a18)
0.1032	Visible whiteness (a12)
0.1024	Red eye (a5)
0.1013	Cornea increased in size (a6)
0.0952	Diabetes (a19)
0.0915	Mass pain (a13)
0.0875	Excessive dryness (a4)
0.0826	Vomiting (a14)
0.0773	Presents of light when eyelid close (a16)
0.0616	Family history of similar disease (a17)
0.0218	Trouble with glasses (a10)

**Table 6 bioengineering-10-00025-t006:** Ranking of Features from PCA-based FS method.

Ranking Score	Attributes (Attribute Number)
0.7619	Cloudy, blurry or foggy vision (a1)
0.5981	Injury to the eye (a3)
0.5054	Excessive dryness (a4)
0.4171	Presents of light when eyelid close (a16)
0.365	Trouble with glasses (a10)
0.3175	Pressure in eye (a2)
0.2719	40+ Age (a18)
0.2306	Family history of similar disease (a17)
0.1976	Vomiting (a14)
0.1678	Mass pain (a13)
0.1389	Red eye (a5)
0.1136	Double vision (a8)
0.09	Cornea increased in size (a6)
0.0719	Problem in identifying color (a7)
0.0549	Myopia (a9)
0.0388	Hard to see in the dark (a11)

**Table 7 bioengineering-10-00025-t007:** Performance of ML models in experiment-2 (Splitting + No FS Applied).

Model	66–34% Split				75–25% Split				80–20% Split			
**Name**	**PR**	**SEN**	**F1-Score**	**ACC**	**PR**	**SEN**	**F1-Score**	**ACC**	**PR**	**SEN**	**F1-Score**	**ACC**
DT	0.97	0.97	0.97	96.875%	0.97	0.97	0.97	97.163%	0.97	0.96	0.96	96.460%
NB	0.95	0.94	0.94	94.271%	0.96	0.95	0.95	95.035%	0.95	0.94	0.94	93.805%
RF	1.00	0.81	0.88	81.25%	1.00	0.82	0.89	81.56%	1.00	0.78	0.87	77.876%
AB	0.98	0.98	0.98	97.51%	0.98	0.98	0.98	98.05%	0.98	0.98	0.98	97.69%
LR	0.98	0.98	0.98	97.917%	0.99	0.99	0.99	**98.582%**	0.97	0.97	0.97	97.345%
k-NN	0.97	0.97	0.97	96.875%	0.97	0.97	0.97	97.163%	0.96	0.96	0.96	96.46%
Bagging	0.91	0.91	0.91	91.513%	0.91	0.84	0.82	91.56%	0.80	0.90	0.89	90.088%
XGBoost	0.98	0.98	0.98	**98.579%**	0.99	0.99	0.99	98.581%	0.98	0.98	0.98	**98.23%**
SVM	0.64	0.74	0.65	74.479%	0.63	0.74	0.64	73.759%	0.59	0.70	0.59	69.912%

**Table 8 bioengineering-10-00025-t008:** Performance of ML models in experiment-3 (Cross Validation + FS Applied).

Cross Fold Validation with FS Method													
**Methods**	**Feature Selection Methods**	**3-fold**				**5-fold**				**10-fold**			
		P	R	F1	ACC	P	R	F1	ACC	P	R	F1	ACC
DT	PC	0.914	0.892	0.892	89.17%	0.918	0.899	0.900	89.88%	0.913	0.897	0.896	89.70%
	IG	0.902	0.893	0.896	89.34%	0.906	0.897	0.899	89.70%	0.907	0.899	0.899	89.88%
	PCA	0.925	0.911	0.912	91.12%	0.926	0.913	0.914	91.30%	0.926	0.913	0.914	91.30%
	Relief	0.916	0.899	0.899	89.88%	0.921	0.899	0.899	89.88%	0.918	0.899	0.899	89.88%
	All *	0.960	0.950	0.950	94.85%	0.960	0.960	0.960	96.81%	0.930	0.930	0.920	96.98%
NB	PC	0.920	0.909	0.907	90.94%	0.923	0.909	0.907	90.94%	0.922	0.909	0.907	90.94%
	IG	0.924	0.915	0.913	91.47%	0.910	0.909	0.909	90.94%	0.898	0.899	0.898	89.88%
	PCA	0.924	0.911	0.913	91.12%	0.925	0.911	0.913	91.12%	0.926	0.911	0.913	91.12%
	Relief	0.916	0.904	0.902	90.41%	0.912	0.901	0.899	90.05%	0.916	0.902	0.901	90.23%
	All *	0.960	0.960	0.960	95.56%	0.960	0.960	0.950	95.92%	0.950	0.930	0.390	95.74%
RF	PC	0.920	0.906	0.905	90.59%	0.923	0.911	0.910	91.12%	0.919	0.908	0.907	90.76%
	IG	0.908	0.902	0.901	90.23%	0.923	0.915	0.915	91.47%	0.918	0.909	0.909	90.94%
	PCA	0.911	0.899	0.901	89.88%	0.895	0.890	0.890	88.99%	0.899	0.892	0.893	89.17%
	Relief	0.923	0.904	0.902	90.41%	0.924	0.904	0.904	90.41%	0.923	0.904	0.903	90.41%
	All *	0.990	0.990	0.990	98.40%	1.000	1.000	1.000	98.58%	1.000	1.000	1.000	97.87%
AB	PC	0.913	0.901	0.900	90.05%	0.923	0.908	0.908	90.80%	0.921	0.908	0.907	90.76%
	IG	0.911	0.902	0.901	90.23%	0.913	0.904	0.905	90.41%	0.925	0.913	0.913	91.30%
	PCA	0.895	0.890	0.890	88.99%	0.909	0.901	0.901	90.05%	0.905	0.897	0.898	89.70%
	Relief	0.920	0.902	0.901	90.23%	0.926	0.909	0.908	90.94%	0.920	0.904	0.903	90.41%
	All *	0.975	0.975	0.975	97.51%	0.980	0.980	0.980	98.05%	0.977	0.977	0.977	97.69%
LR	PC	0.919	0.902	0.904	90.23%	0.924	0.911	0.911	91.12%	0.922	0.909	0.909	90.94%
	IG	0.921	0.909	0.910	90.94%	0.913	0.908	0.908	90.76%	0.927	0.917	0.917	91.65%
	PCA	0.900	0.893	0.895	89.34%	0.904	0.899	0.900	89.88%	0.911	0.902	0.904	90.23%
	Relief	0.927	0.911	0.912	91.12%	0.926	0.909	0.909	90.94%	0.930	0.913	0.913	91.30%
	All *	0.990	0.980	0.950	98.58%	1.000	1.000	1.000	98.94%	1.000	1.000	1.000	98.94%
k-NN	PC	0.930	0.915	0.913	91.47%	0.929	0.917	0.915	91.65%	0.926	0.915	0.912	91.47%
	IG	0.908	0.908	0.907	90.76%	0.912	0.906	0.907	90.59%	0.926	0.915	0.915	91.47%
	PCA	0.892	0.886	0.887	88.63%	0.906	0.899	0.899	89.88%	0.901	0.895	0.896	89.52%
	Relief	0.929	0.908	0.908	90.76%	0.925	0.906	0.906	90.59%	0.922	0.902	0.902	90.23%
	All *	0.960	0.960	0.950	96.45%	0.980	0.970	0.970	96.27%	1.000	1.000	1.000	96.63%
Bagging	PC	0.914	0.897	0.896	89.70%	0.926	0.909	0.907	90.94%	0.925	0.911	0.910	91.00%
	IG	0.909	0.892	0.889	89.17%	0.898	0.897	0.894	89.70%	0.897	0.895	0.893	89.30%
	PCA	0.911	0.899	0.899	89.88%	0.909	0.895	0.895	89.52%	0.905	0.890	0.889	88.99%
	Relief	0.905	0.885	0.885	88.45%	0.911	0.897	0.893	89.70%	0.914	0.897	0.896	89.70%
	All *	0.930	0.940	0.970	95.06%	0.940	0.960	0.950	95.58%	0.940	0.950	0.890	95.58%
XGBoost	PC	0.920	0.904	0.905	90.41%	0.926	0.911	0.912	91.12%	0.926	0.911	0.911	91.12%
	IG	0.929	0.917	0.917	91.65%	0.920	0.911	0.913	91.12%	0.924	0.913	0.915	91.30%
	PCA	0.916	0.906	0.907	90.59%	0.915	0.906	0.907	90.59%	0.918	0.908	0.909	90.76%
	Relief	0.929	0.913	0.913	91.30%	0.923	0.908	0.908	90.76%	0.927	0.909	0.910	90.94%
	All *	0.980	0.980	0.980	98.58%	0.990	0.990	0.990	98.58%	0.980	0.980	0.980	98.05%
SVM	PC	0.934	0.911	0.907	91.12%	0.935	0.913	0.907	91.30%	0.928	0.911	0.905	91.12%
	IG	0.917	0.909	0.911	90.94%	0.913	0.906	0.907	90.59%	0.898	0.890	0.891	88.99%
	PCA	0.904	0.888	0.888	88.81%	0.902	0.890	0.888	88.99%	0.899	0.892	0.891	89.17%
	Relief	0.935	0.911	0.909	91.12%	0.932	0.908	0.906	90.76%	0.931	0.908	0.906	90.76%
	All *	0.980	0.980	0.980	98.76%	1.000	1.000	1.000	98.94%	1.000	1.000	1.000	99.11%

* All features, P: Precision, R: Recall, F1: F1-score, ACC: Accuracy.

**Table 9 bioengineering-10-00025-t009:** Performance of ML models in experiment-4 (Cross-validation + No FS Applied).

Model	3-fold				5-fold				10-fold			
**Name**	**PR**	**SEN**	**F1-Score**	**ACC**	**PR**	**SEN**	**F1-Score**	**ACC**	**PR**	**SEN**	**F1-Score**	**ACC**
DT	0.96	0.95	0.95	94.850%	0.96	0.96	0.96	96.805%	0.93	0.93	0.92	96.980%
NB	0.96	0.96	0.96	95.561%	0.96	0.96	0.95	95.915%	0.95	0.93	0.93	95.742%
RF	0.99	0.99	0.99	98.402%	1.00	1.00	1.00	98.581%	1.00	1.00	1.00	97.870%
AB	0.98	0.98	0.98	97.510%	0.98	0.98	0.98	98.050%	0.98	0.98	0.98	97.690%
LR	0.99	0.98	0.95	98.579%	1.00	1.00	1.00	98.936%	1.00	1.00	1.00	98.938%
k-NN	0.96	0.96	0.95	96.447%	0.98	0.97	0.97	96.271%	1.00	1.00	1.00	96.626%
Bagging	0.93	0.94	0.97	95.062%	0.94	0.96	0.95	95.579%	0.94	0.95	0.89	95.578%
XGBoost	0.98	0.98	0.98	98.579%	0.99	0.99	0.99	98.581%	0.98	0.98	0.98	98.051%
SVM	0.98	0.98	0.98	98.756%	1.00	1.00	1.00	98.936%	1.00	1.00	1.00	**99.110%**

**Table 10 bioengineering-10-00025-t010:** Comparison with existing works.

Existing Works	Features Used	Feature Selection Used	Methods	Evaluation
Nazir et al. [19]	Extracted features using Densenet-100	No	Improved CenterNet	Accuracy (using Aptos-2019 dataset: 97.93%, using IDrID dataset: 98.10%)
Bodapati et al. [20]	Feature fusion	No	Deep neural Network	Accuracy (using Aptos-2019 dataset: 84.31%)
Khan et al. [21]	Manual extracted retinal features	No	CNN with VGG-19	Accuracy (97.47%)
Sarki et al. [22]	None	No	CNN with RMSprop Optmizer	Accuracy (81.33%)
Pahuja et al. [23]	None	No	SVM and CNN	Accuracy (SVM:87.5% and CNN: 85.42%)
Malik et al. [24]	None	No	DT, NB, RF and NN	Accuracy (RF: 86.63%)
Proposed	None	Yes (PC, IG, PCA, Relief)	DT, NB, RF, AB, LR, k-NN, Bagging, XGBoost and SVM	Accuracy (Highest Accuracy Obtained: 99.11% (SVM))

## Data Availability

Data are available from the corresponding author and can be shared with anyone upon reasonable request.

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
