# Peer review of "An Efficient Approach to Predict Eye Diseases from Symptoms Using Machine Learning and Ranker-Based Feature Selection Methods"

_bioengineering, 2022, doi:10.3390/bioengineering10010025_

Round 1

Reviewer 1 Report

The paper describes a solution for the prediction of eye diseases based on given symptoms. The idea is based on feature selection with k-fold cross-validation and then applying a machine learning model. In my opinion, the paper needs some improvements:

1) Explain the novelty of your proposal. It seems to me that an idea is a well-known approach. However, the used tools are not modified at all.

2) The proposal describes mainly known solutions and tools.

3) Experimental section is based on the basic database without proper comparison with state of art, or even modified used tools that were published in the last 2-3 years.

4) The background of the paper is based mainly on outdated research. The authors should describe their solution according to the last 2 years mainly.

Author Response

Firstly, thank you so much for your effort and time to review this paper. We appreciate your contribution and guidance for the paper.

Thanks for choosing “English language and Style are fine/minor spell check required”. We have gone through the paper for spell checking and language has been modified where necessary. Based on the valuable comments, we have updated the paper accordingly.

  1. Explain the novelty of your proposal. It seems to me that an idea is a well-known approach. However, the used tools are not modified at all.

Response: The novelty of the proposed approach is listed in the introduction section. With the new benchmark dataset that we have proposed, firstly we have experimented over the state-of-the-art classification algorithms (referred as tools). The experimentations have been performed with different setups (i.e., data splitting for training-testing and cross-validation).

  1. The proposal describes mainly known solutions and tools.

Response: In this research, we aimed to find out the most efficient state-of-the-art classifier for the dataset we proposed. This research is going to set a background for the future scopes of applying modified tools.

  1. Experimental section is based on the basic database without proper comparison with state of art, or even modified used tools that were published in the last 2-3 years.

Response: A comparison table (Table-10) added in the experimentation section to compare the existing works with the proposed approach. Most of the existing works are recent and within 2-3 years.  

  1. The background of the paper is based mainly on outdated research. The authors should describe their solution according to the last 2 years mainly.

Response: The related works section has been updated with some recent works of last two years.

Reviewer 2 Report

1.Authors can add the strengths (positive points) and shortcomings of the study.

2.Conclusion of the study can be better crafted by revisiting the critical points of the study like accuracy and relevance of the results.

Author Response

Thank you so much for your effort and time to review this paper. We appreciate your contribution and guidance for the paper. Based on the valuable comments, we have updated the paper accordingly.

  1. Authors can add the strengths (positive points) and shortcomings of the study.

Response: The positive points (contributions) are listed in introduction section and some of the shortcomings with some future directions are mentioned in the conclusion section.

  1. Conclusion of the study can be better crafted by revisiting the critical points of the study like accuracy and relevance of the results

Response: The conclusion section is revised according to the comments.

Reviewer 3 Report

What factors deserve the most attention because they affect the organ's vitality?

What are the most significant discrepancies, if any, between the analysis results and the comparative study of the methods?

When it came to classifying eye diseases, did Malik et al. take a data-driven approach? When putting together their dataset, Malik et al.

A comparison with recent studies and methods would be appreciated.

The conclusion should state the scope for future work.

The comparison of different methods using clear graphs should be explained.

Discuss the plans concerning the research state of progress and its limitations.

An error and statistical analysis of data should be performed.

A detailed explanation of the analysis/processing steps.

The abstract should clarify what is precisely proposed (the technical contribution) and validate the proposed approach.

Literature review techniques should include the current system's issues and how the author proposes to overcome the same.

Add the chart for the given process with a description.

The paper does not clearly explain its advantages concerning the literature: it is not clear the novelty and contributions of the proposed work: does it offer a new method? Or does the innovation only consists of the application?

The advantage of the proposed method concerning other ways in the literature should be clarified.

Author Response

Thank you so much for your effort and time to review this paper. We appreciate your contribution and guidance for the paper. Based on the valuable comments, we have updated the paper accordingly.

  1. What factors deserve the most attention because they affect the organ's vitality?

Response: The common features extracted from our investigation are attributes a1, a2 and a3. These attributes are the main factors for all the five eye disease we have worked on. This is also explained in the paper.

  1. What are the most significant discrepancies, if any, between the analysis results and the comparative study of the methods?

Response: A new section is added for comparing the existing work with the proposed approach. An analysis has been added.

  1. When it came to classifying eye diseases, did Malik et al. take a data-driven approach? When putting together their dataset, Malik et al. A comparison with recent studies and methods would be appreciated.

Response: The proposed work has been compared with Malik et. al. in Table 10.

  1. The conclusion should state the scope for future work.

Response: Future scopes are added in the conclusion.

  1. The comparison of different methods using clear graphs should be explained.

Response: Because of the restriction of dimensionality, we have shown some of the results in form of graphs and rest of the results are compared using tables. The nature of the experimentations with different setups it is quite difficult to show everything in a graph. Therefore, ample amount of explanation has been provided with the tables.

  1. Discuss the plans concerning the research state of progress and its limitations.

Response: The limitations of the proposed work along with some future prospects are mentioned in the conclusion section.

  1. An error and statistical analysis of data should be performed.

Response: The statistical property of the dataset is provided in table 1.

  1. A detailed explanation of the analysis/processing steps.

Response: An overview of the proposed approach has been conceptualized and presented in figure 1. All the steps of the figure are explained in detail in the research methodology section with subsections.

  1. The abstract should clarify what is precisely proposed (the technical contribution) and validate the proposed approach.

Response:  Thanks for pointing out. The abstract is revised accordingly.

  1. Literature review techniques should include the current system's issues and how the author proposes to overcome the same.

Response: The related works section has been updated with some recent works of last two years.

  1. Add the chart for the given process with a description.

Response: The overall research methodology has been depicted in a block diagram in figure 1. The detailed description of the process is also added in section 4.

  1. The paper does not clearly explain its advantages concerning the literature: it is not clear the novelty and contributions of the proposed work: does it offer a new method? Or does the innovation only consist of the application?

Response: In this research, we aimed to find out the most efficient state-of-the-art classifier for the dataset we proposed. This research is going to set a background for the future scopes of applying modified tools.

  1. The advantage of the proposed method concerning other ways in the literature should be clarified.

Response: Thanks for pointing out. The related work section has been revised accordingly.

Round 2

Reviewer 1 Report

The paper was corrected and improved.